# AIA: LEARN TO DESIGN GREEDY ALGORITHM FOR NP-COMPLETE PROBLEMS USING NEURAL NETWORKS

## ABSTRACT

Algorithm design is an art that heavily requires intuition and expertise of the human designers as well as insights into the problems under consideration. In particular, the design of greedy-selection rules, the core of greedy algorithms, is usually a great challenge to designer: it is relatively easy to understand a greedy algorithm while it is always difficult to find out an effective greedy-selection rule. In the study, we present an approach, called AIA, to learn algorithm design with the aid of neural networks. We consider the minimum weighted set cover problem (WSCP), one of the NP-hard problems, as an representative example. Initially, we formulate a given WSCP as an 0-1 integer linear program (ILP): each variable $x_i$ has two options, i.e., $x_i = 0$, which denotes abandon of the set $s_i$, and $x_i = 1$, which denotes selection of $s_i$. Each option of a variable leads to a sub-problem with respect to the original ILP problem. Next, we design a generic search framework to find the optimal solution to the ILP problem. At each search step, the value of a variable is determined with the aid of neural networks. The key of our neural network is the loss function: the original ILP problem and the sub-problems generated by assigning a variable $x_i$ should satisfy the Bellman-Ford equation, which enables us to set the dissatisfication of Bellman-Ford equation as loss function of our neural network. The neural network is used as greedy-selection rule. Experimental results on representative instances suggest that using the NN-based greedy selection rule, we can successfully find the optimal solutions. More importantly, the NN-based greedy-selection rule outperform the outstanding Chavatal greedy algorithm, which was designed by human expert. The basic idea of our approach can be readily extended without significant modification to design greedy algorithm for other NP-hard problems.

## 1 INTRODUCTION

NP-complete problems, the hardest ones in the NP class, can be validated in polynomial time, but no polynomial-time algorithm has yet been found to solve these problems. A great variety of practical problems can be formulated as NP-complete problems, such as strategic planning, production planning, facility location problems, as well as a variety of scheduling and routing problems. Thus, despite the hardness of these problems, designing efficient solving algorithms for NP-complete problems is highly desired.

The weighted set cover problem (WSCP) is a classical NP-complete problem, which aims to find a subset of columns that cover all the rows of a 0-1 matrix at minimal cost (Karp, 1972). The algorithms to solve WSCP, say branch-and-bound and branch-and-cut, can only handle instances with limited size. Therefore, considerable efforts have been devoted to design heuristics and meta-heuristics that can find optimal or near optimal solutions to large-scale WSCP problems within a reasonable time. The latest works on meta-heuristic approaches for the WSCP include genetic algorithms, ant colony optimization, simulated annealing, tabu Search.

With the breakthrough of deep learning (DL) in solving practical problems, many researchers try to use DL to solve combinatorial optimization problems. Training a neural network end-to-end with supervised learning to solve theoretically complex combinatorial optimization problems is a difficult problem. On the one hand, traditional algorithms and mathematical methods have a relatively complex theoretical foundation while the neural network as a black box lacks theoretical foundation.

On the other hand, many practical problems have their specific data characteristics while DL models usually require a large amount of labeled data under the distribution and constructing labeled data needs to know the optimal solution of the original problem, so it is very difficult to build a large-scale dataset like ImageNet(Russakovsky et al., 2015).

Thus, we are not directly solving this problem end-to-end like other fields. In the study, we present an approach, called AIA, to learn algorithm design with the aid of neural networks. The specific goal of this paper is to use machine learning to find greedy rules for solving WSCP and then design an efficient and practical algorithm for WSCP.

Our main contributions of this work are as follows:

1. We propose an idea to solve the problem, where the neural network learns greedy strategies to assist researchers in designing algorithms instead of using deep learning to solve the problem end-to-end.

2. We propose the NNVal algorithm, which uses a simple neural network to score recursive sub-problems that are used to guide a multi-step decision-making process, and a special novel loss function is designed to train the neural network.

3. We propose the NNGreedy algorithm for WSCP. The experimental results on multiple datasets show that compared with greedy algorithms designed based on human experience, such as the Chvatal algorithm, the NNGreedy algorithm can obtain better solution.

## 2 RELATED WORKS AND BACKGROUND

There exist two traditional approaches to solve combinatorial optimization problems: exact algorithms and approximate/heuristic algorithms. Exact algorithms are guaranteed to find optimal solutions, but they become intractable when the problem scales up. Approximate algorithms trade optimality for computational efficiency. They are problem-specific, often designed by iteratively applying a simple man-crafted rule, known as heuristic. Their complexity is polynomial and their quality depends on an approximate ratio that characterizes the worst/average-case error w.r.t the optimal solution.

For NP-complete or even NP-hard constraint programming problems, the exact algorithm usually adopts the divide-and-conquer approach, dividing the solution process into multiple parts, and gradually eliminate poor choices until the optimal solution is found. These methods are essentially exhaustive search and their time complexity are exponential.

The cores of the divide-and-conquer strategy is how to decompose it into sub-problems and which sub-problem to choose to solve first. Taking the Traveling Salesman Problem (TSP) as an example, the solution software Concorde(Applegate et al., 2002) adopts the branch-and-bound combined with cut plane method. For more general mixed integer linear programming problems, the mainstream solvers, for example, Gurobi(Bixby, 2007), CPLEX(Cplex, 2009), XPRESS(Laundy et al., 2009), SCIP(Gamrath et al., 2016), etc., also use branch and bound combined with cutting plane and column generation technology.

When searching for the optimal solution, heuristic rules play an important role. Taking mixed integer programming as an example, in the branch-and-bound process, selecting an appropriate branch variable requires heuristic rules, and each time a sub-problem is re-selected also requires heuristic rules. Choosing appropriate branching variables and subproblems can significantly reduce the search space. In addition, to obtain a good feasible solution as soon as possible to speed up pruning, the algorithm performs a simple and fast primitive heuristic strategy on each subproblem. A good original heuristic can find better feasible solutions faster.

Heuristic rules depend on the specific problem and solution process. In other words, different heuristic rules and different parameters have different effects on different data and at different stages of solution. Therefore, data-driven machine learning technique is a potential heuristic rule design method.

In the last decade, DL has significantly improved Computer Vision, Natural Language Processing and Speech Recognition by replacing hand-crafted features with features learned from data(LeCun et al., 2015). On the one hand, combinatorial optimization algorithms are often used as a com-

plement to deep learning solutions. DETR(Carion et al., 2020) uses a bipartite graph matching algorithm to replace the NMS post-processing in traditional object detection, which solves a pain point in this field. DeepSORT(Wojke et al., 2017) uses the Hungarian algorithm to tell if an object in current frame is the same as the one in previous frame, which is one of the most popular and general object tracking algorithms.

On the other hand, more and more researchers have introduced neural network into Combinatorial Optimization, called Neural Combinatorial Optimization(Garmendia et al., 2022), which attempts to learn good heuristics for solving a set of problems using Neural Network models. The use of machine learning techniques in NP-complete problem solving can be divided into two categories: one is learning from expert knowledge, that is, supervised learning, and the other is learning from experience, that is, reinforcement learning, which is briefly described as follows:

**Supervised learning** is relatively common and easy to implement. In the branch and bound framework of Mixed Integer Programming Solver, there are more than 10 heuristic rules for the selection of branch variables. The Strong Branch strategy is recognized as the branch variable selection strategy that minimizes the size of the branch search tree (Achterberg et al., 2005). It selects the variable branch that minimizes the lower bound of the new sub-problem after the branch every time. The cost is that for each branch, the lower bound of the new subproblem must be calculated. This step introduces a lot of calculations, and absolutely most calculations cannot be reused in subsequent solutions. This makes the strong branching strategy lag behind in solving time compared to other rules. In order to overcome the shortcoming of the strong branch strategy, Marcos Alvarez(Alvarez et al., 2014) used a special kind of decision tree to learn the branch variable selection strategy of Strong Branch. Khalil(Khalil et al., 2016) proposes an instance-specific learning framework. For each problem, the choice of strong branching strategy is recorded on several sub-problems at the beginning, and the features of each variable and each step are extracted at the same time. They train an SVMrank and use the learned model on the following sub-problems. Gasse uses a Graph Convolutional Neural Network(Gasse et al., 2019) to extract deeper information on variables and constraints before each branch to learn the choice of strong branching strategies. He(He et al., 2014) designed a machine learning algorithm,in which among all open subproblems, the subproblem whose subtree contains the optimal solution will be selected. Chaitanya's work(Joshi et al., 2019) uses a deep graph network by supervision to predict the probabilities of an edge to be in the TSP tour, which is more sample efficient compared to reinforcement learning, whose feasible tour is generated by beam search.

**Reinforcement learning** techniques are mostly used in solving algorithms for specific NP-hard problems. Taking the TSP problem as an example, Vinyals(Vinyals et al., 2015) proposed a pointer network, and introduced a pointer in the decoder RNN, which solves the problem of variable size output dictionaries using the mechanism of neural attention. This model solves the problem that the output scale strictly depends on the input scale during training. Bello(Bello et al., 2016) uses a similar model structure, sets the current total distance as a reward, and uses reinforcement learning to train the network, which solves parts of problems that supervised learning is difficult to deal with, such as non-unique standard answers. Kool and Welling(Kool et al., 2018) replaced the Recurrent Neural Network (RNN) with a Graph Neural Network (GNN) to process the input. All three above use reinforcement learning to train end-to-end models. Astounding results from Transformer(Vaswani et al., 2017) models on NLP and CV tasks(Khan et al., 2021) have intrigued the researcher to study their application to TSP. Xavier and Thomas(Bresson & Laurent, 2021) propose to adapt the Transformer architecture to the combinatorial TSP. Training is done by reinforcement learning, hence without TSP training solutions, and decoding uses beam search. As for the learning of specific heuristic rules, Khalil uses the graph neural network (Khalil et al., 2017) to encode graph information to make choices.

In general, reinforcement learning methods are more widely used in solving combinatorial optimization problems, which don't require too much prior knowledge, while supervised learning methods are more efficient in sampling and training. All in all, the use of machine learning to assist the solution of combinatorial optimization problems is a very promising research trend. The main question is whether DL can learn better heuristics from data, i.e. replacing human-designed heuristics.

## 3 STATE-TRANSITION EQUATION

The WSCP is the problem of covering the rows of an $m$-row, $n$-column, 0-1 matrix $\boldsymbol{A}$ with a subset of the columns at minimum cost. Problem 1 is a matrix form of WSCP.

$$
\min \ z = \boldsymbol{c}^T \boldsymbol{x}
$$
$$
s.t. \quad \begin{cases} \boldsymbol{A}\boldsymbol{x} \geq \boldsymbol{b} \\ x_j = 0 \ or \ 1 \quad (j = 1, 2, \cdots, n). \end{cases} \tag{1}
$$

We define a state $s_k = [\boldsymbol{A}, \boldsymbol{b}, \boldsymbol{c}; k]$ $(k = 0, 1, ..., n)$. $k$=0 represents the original programming problem and $k > 0$ means that the variables $x_1, x_2, ..., x_k$ have already been fixed. We define the problem as a multistep decision-making process, determining whether a variable $x_k$ is 0 or 1 at each step. If $x_1$=1 at the first step, $\boldsymbol{A}$ with m rows and n columns turns to be $\boldsymbol{A}'$, which has m rows and $n - 1$ columns and $\boldsymbol{b}$ turns to be $\boldsymbol{b} - \alpha_1$ ($\alpha_j$ is the $j$-th column of $\boldsymbol{A}$). We redefine the problem as follows,

$$
\min \ z = \boldsymbol{c}^T \boldsymbol{x}
$$
$$
s.t. \quad \begin{cases} \boldsymbol{A}\boldsymbol{x} \geq \boldsymbol{b} \\ x_1, x_2, ..., x_k \ are \ fixed \\ x_{k+1}, x_{k+2}, ..., x_n = 0 \ or \ 1 \end{cases} \tag{2}
$$

We define $f(s_k)$ as the optimal solution of the subprogram 2 under the state $s_k$. Therefore state-transition equation can be obtained,

$$
f(s_{k+1}) = \min_{x_{k+1} = 0 \ or \ 1} f(s_k), \quad k = 0, ..., n - 1 \tag{3}
$$

## 4 METHOD

We use a neural network to score each sub-problem and guide multi-step decision-making according to equation 3 instead of predicting the solution to the sub-problem. We call this algorithm NNVal.[1]

### 4.1 PROBLEM REFORMULATION

Supposing the optimal solution of the sub-problem 2 is $f(\boldsymbol{A}, \boldsymbol{b}, \boldsymbol{c})$, jwe have the following state-transition equation at state $s_k$,

$$
f(\boldsymbol{A}', \boldsymbol{b}', \boldsymbol{c}') = \min \left( f(\boldsymbol{A}'', \boldsymbol{b}', \boldsymbol{c}''), \quad \boldsymbol{c}_{k+1} + f(\boldsymbol{A}'', \boldsymbol{b}' - \alpha_{k+1}, \boldsymbol{c}'') \right) \tag{4}
$$

$$
\begin{aligned}
\boldsymbol{A}' &= \boldsymbol{A}'_{m \times (n-k)} &= (\alpha_{k+1}, \alpha_{k+2}, ..., \alpha_n) \\
\boldsymbol{A}'' &= \boldsymbol{A}''_{m \times (n-k-1)} &= (\alpha_{k+2}, \alpha_{k+3}, ..., \alpha_n) \\
\boldsymbol{b}' &= \boldsymbol{b}'_m &= \boldsymbol{b} - (x_1^* \alpha_1 + ... + x_k^* \alpha_k) \\
\boldsymbol{c}' &= \boldsymbol{c}'_{n-k} &= (c_{k+1}; c_{k+2}; ...; c_n) \\
\boldsymbol{c}'' &= \boldsymbol{c}''_{n-k-1} &= (c_{k+2}; c_{k+3}; ...; c_n)
\end{aligned}
$$

where, $x_k^*$ is the variable determined at state $s_{k-1}$ and $\alpha_k$ represents the $k$-th column of $\boldsymbol{A}$. We define $f(\boldsymbol{A}, \boldsymbol{b}, \boldsymbol{c}) = +\infty$ when $\boldsymbol{A}\boldsymbol{x} \geq \boldsymbol{b}$ has no feasible solution.

Thus, given the function $f$, it is easy to get:

$$
x_{k+1} = \begin{cases} 0 & f(\boldsymbol{A}'', \boldsymbol{b}', \boldsymbol{c}'') \leq c_{k+1} + f(\boldsymbol{A}'', \boldsymbol{b}' - \alpha_{k+1}, \boldsymbol{c}'') \\ 1 & f(\boldsymbol{A}'', \boldsymbol{b}', \boldsymbol{c}'') > c_{k+1} + f(\boldsymbol{A}'', \boldsymbol{b}' - \alpha_{k+1}, \boldsymbol{c}'') \end{cases} \tag{5}
$$

---

[1]Appendix B provide the training process including flowchat and pseudocode

## 4.2 NNVAL ALGORITHM

A simple idea is to fit $f$ with a neural network, but it may not be feasible. On the one hand, it is due to the difficulty of obtaining labeled data, and on the other hand, due to the characteristics of the neural network itself, the input changes little, but the output may change greatly, so it is difficult to accurately predict the target. We decide to make full use of the state-transition equation. Our model $g_\theta$ does not directly learn the optimized values of the sub-problems, but learns the recursive relationship between the original problem and the sub-problems.

For preserving all information of the problem and simplifying the training procedure, we use a simple two-layer fully-connected neural network to learn the recursive relationship. The model $g_\theta$ input is the same as the input of $f$, whose input layer has $m \times n + m + n$ nodes, corresponding to $A$, $b$ and $c$ of the original Problem 1. As for sub-problem input, $\alpha_k$ and $c_k$ corresponding to fixed variables $x_k$ are set to 0. The output layer of the network has 1 node. 1 hidden layer is in the middle, which has $m + n$ nodes. The whole model is shown in Figure 1.

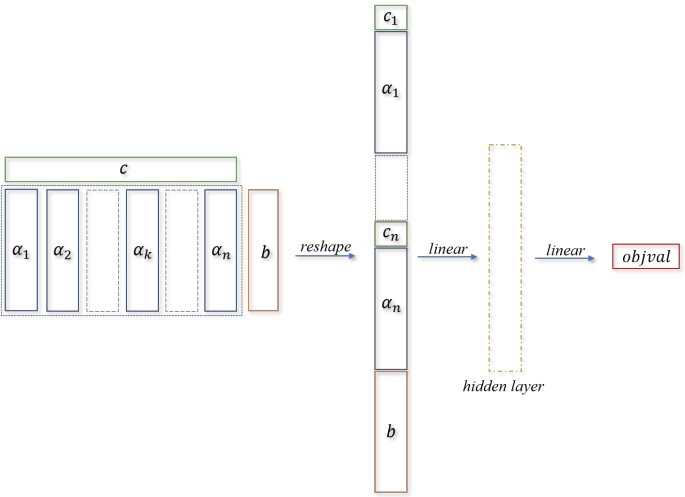

Figure 1: Neural Network for NNVal

## 4.3 LOSS FUNCTION

We define the state transition loss function for one-step decision-making just as follows,

$$\mathcal{L}(P;\theta) = L_{square} + L_{aux} \tag{6}$$

In dynamic programming, the recurrence relation is as follow,

$$\text{Objval}(P) = \min\{\text{Objval}(SP_0), \text{Objval}(SP_1)\} \tag{7}$$

where $P$ is the original problem and $SP$ is sub-problem of $P$. We hope our model can also learn the recursive relationship, so we define $L_{square}$ as follows,

$$L_{square} = (g_\theta(P) - min\{g_\theta(SP_0), g_\theta(SP_1)\})^2 \tag{8}$$

We want the output of the original problem to be as close as possible to the output of the correct sub-problem. However, this is not enough, because sometimes there are situations where sub-problems are not feasible. Thus, we add the auxiliary item $L_{aux}$ as follow,

$$L_{aux} = \text{ReLU}\,(g_\theta(\text{feasible SP}) - g_\theta(\text{infeasible SP})) \tag{9}$$

where $\text{ReLU}(x) = \max(x, 0)$. We know $L_{aux}$ is equal to 0 when the model output of feasible sub-problem is less than the infeasible, and if infeasible sub-problem is selected, this item will generate a positive penalty. All in all, $L_{square}$ is mainly used to make model learn the recursive relationship, and $L_{aux}$ is used to control the case where the sub-problem is not feasible.[2]

### 4.4 IMPROVEMENT

The algorithm discussed before relies on external library functions to judge the feasibility of the sub-problem, so it is suitable for any 0-1 ILP. This section attempts to mine the nature of the weighted set coverage problem itself, which is more controllable and flexible. In order to facilitate the distinction, we call the algorithm discussed before as NNVal V1, and the algorithm to be discussed next is called NNVal V2.

In NNVal V2, we still use a two-layer fully-connected neural network just as V1. The difference of model from V1 is that a ReLU layer is added. The structure of the neural network is shown in Figure 2, where $\tilde{b} = \text{ReLU}(b)$.

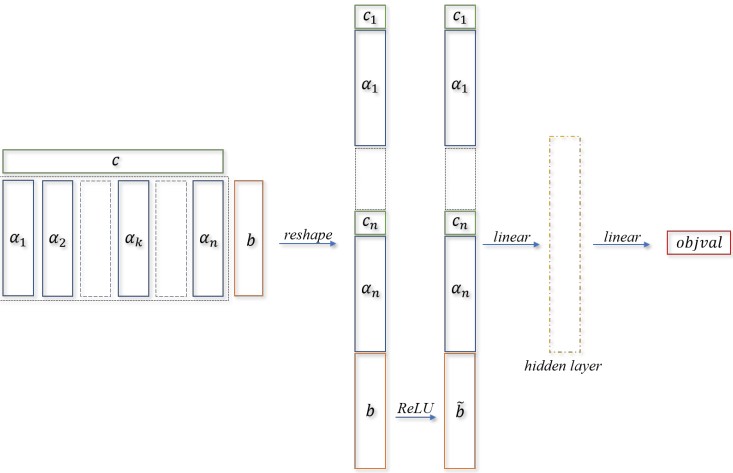

Figure 2: Neural network for improved version of NNVal

Let's rethink the sub-problem corresponding to state $s_k = [\boldsymbol{A}, \boldsymbol{b}, \boldsymbol{c}; k]$,

$$
\begin{aligned}
\min \quad & c_{k+1}x_{k+1} + ... + c_n x_n + (c_1 x_1^* + ... + c_k x_k^*) \\
\text{s.t.} \quad & \alpha_{k+1}x_{k+1} + ... + \alpha_n x_n \geq b - (\alpha_1 x_1^* + ... + \alpha_k x_k^*) \\
& x_{k+1}, x_{k+2}, ..., x_n = 0 \ or \ 1
\end{aligned}
\tag{10}
$$

where the constant term on the right side of the constraint inequality takes a integer value not exceeding 1. In fact, for such a 0-1 ILP, the value of $b - (\alpha_1 x_1^* + ... + \alpha_k x_k^*)$ being 0 or less than 0 has no effect on the solution, so we can use ReLU to reduce the complexity of training. The $i$-th row of the inequality constraint can be written as,

$$
a_{i,k+1}x_{k+1} + ... + a_{i,n}x_n \geq b_{i,k+1}
\tag{11}
$$

where $b_{i,k+1} \in \mathbb{Z} \cap \{x \mid x \leq 1\}$ is the $i$-th row element of $b - (\alpha_1 x_1^* + ... + \alpha_k x_k^*)$, $a_{i,j} \in \{0, 1\}$ is the $i$-th row element of $\alpha_j$ and $x_j \in \{0, 1\}$.

In fact, $x_{k+1}, ..., x_n$ can be arbitrarily 0 or 1 if $b_{i,k+1} \leq 0$. However if $b_{i,k+1} = 1$, then the row constraint is possible to have further restrictions on $x_{k+1}, ..., x_n$ being 0 or 1. In particular, if $b_{i,k+1} = 1$, $a_{i,l} = 1$ and $a_{i,k+1} = ... = a_{i,l-1} = a_{i,l+1}, ... = a_{i,n} = 0$, then the line constraint becomes $x_l \geq 1$ and we get $x_l = 1$.

---

[2]More interpretion is shown in appendix A

Thus, before the state transition, check whether $x_{k+1}$ need to be fixed to 1. If so directly specify $x_{k+1} = 1$, otherwise make a state transition according to the result of the model $g_\theta$.

### 4.5 NNGREEDY ALGORITHM

Based on NNVal algorithm, we design greedy rule and greedy algorithm NNGreedy. The NNGreedy algorithm makes decisions step by step from $x_1$ to $x_n$, and finally obtains a complete solution. Without loss of generality, we record $P_k$ as the original problem of the $k$-th step. $SP_{k0}$ corresponds to the sub-problem when $x_k = 0$ and $SP_{k1}$ corresponds to the sub-problem when $x_k = 1$. The greedy rule adopted by the NNGreedy algorithm is as follows,

---
**Algorithm 1** NNGreedy algorithm greedy rule
---
1: **if** $g_\theta(SP_{k0}) < g_\theta(SP_{k1})$ **then**
2:      $x_k = 0$
3: **else**
4:      $x_k = 1$
5: **end if**

---

In a specific WSCP, for $S_i \in \mathcal{F}, \forall i \in \{1, 2, ..., n\}$, $x_i = 0$ means not choosing $S_i$ and $x_i = 1$ means $S_i$ is selected. The pseudo-code of the NNGreedy algorithm can be expressed as follows,

---
**Algorithm 2** NNGreedy greedy algorithm
---
1: **for** $i = 1 \to n$ **do**
2:      Calculated $x_i$ according to the NNGreedy greedy rule
3: **end for**
4: Output $x_1, x_2, ..., x_n$ and $\sum_{i=1}^n w_i x_i$

---

## 5 EXPERIMENTS

### 5.1 NNVAL EXPERIMENTS

This article uses PyTorch1.6. The CPU is an Intel Core i7-8700K, and the GPU is an NVIDIA GeForce GTX 1080Ti. We randomly generate some weighted set coverage problems as dataset according to the method of Balas & Ho (1980). We use the two scale as follow,

1. Small-scale ILP instance, full set $|U| = 20$, 20 subsets $S_1, S_2, ..., S_{20}$, constraint matrix density is 0.1, that is, the probability of the number 1 appearing in the constraint matrix.

2. Large-scale ILP instance, full set $|U| = 50$, 50 subsets $S_1, S_2, ..., S_{50}$, constraint matrix density is the same as small-scale instance.

The NNVal algorithm is mainly used to assist decision making at every step, and the sub-problems at different stages have different scales. In order to prove the effectiveness of the algorithm, we conduct experiments on problems of different scales. We choose sub-problem pairs $(SP_0, SP_1)$ to evaluate.

Table1 shows the estimation of the order between the sub-problems of NNVal pairs on small-scale and big-scale instances, and their estimation accuracy is 95.4% and 86.5%.

### 5.2 NNGREEDY EXPERIMENTS

We compared the NNgreedy algorithm with the Chvatal's greedy algorithm, which is designed purely based on human experience, in terms of **time and** solution quality

On 100 small-scale instances, we compare the NNGreedy algorithm and the Chvatal greedy algorithm from two perspectives:

Table 1: The estimation relationship on sub-problem pairs in 2 scales

| SCALE | | Obj($SP_0$)<Obj($SP_1$) | Obj($SP_0$)>Obj($SP_1$) | Acc |
|---|---|---|---|---|
| Big | $g_\theta(SP_0)<g_\theta(SP_1)$ | 217 | 139 | 86.50% |
| | $g_\theta(SP_0)>g_\theta(SP_1)$ | 128 | 1494 | |
| Small | $g_\theta(SP_0)<g_\theta(SP_1)$ | 281 | 33 | 95.39% |
| | $g_\theta(SP_0)>g_\theta(SP_1)$ | 28 | 982 | |

Table 2: Comparison of solution quality on small scale instances

| Instance | Optimal Obj | Chvatal Greedy | | NNGreedy | |
|---|---|---|---|---|---|
| | | Objective Value | Gap | Objective Value | Gap |
| I20_1 | 416 | 416 | 0 | 416 | 0 |
| I20_2 | 621 | 676 | 8.86% | **621** | **0** |
| I20_3 | 494 | 598 | 21.05% | **503** | **1.82%** |
| I20_4 | 451 | 490 | 10.87% | **451** | **0** |
| I20_5 | 542 | 542 | 0 | 542 | 0 |
| I20_6 | 520 | 553 | 6.35% | **520** | **0** |
| I20_7 | 570 | 678 | 18.95% | **570** | **0** |
| I20_8 | 602 | 702 | 16.61% | **602** | **0** |
| I20_9 | 529 | 549 | 3.78% | **529** | **0** |
| I20_10 | 427 | **427** | **0** | 432 | 1.17% |

1. Comparison of the number of instances to obtain the optimal solution: NNGreedy obtains the optimal solution on 87 instances, and the Chvatal greedy algorithm obtains the optimal solution on 11 instances.

2. Comparison of solution Quality: On 87 instances, the solution obtained by NNGreedy is better than the solution obtained by Chvatal greedy, and on 3 instances, the solution obtained by NNGreedy is inferior to the Chvatal greedy algorithm.

Table 2 shows the specific solution quality comparison of the two greedy algorithms on 10 small-scale instances. As shown in the table 2, the solution quality of NNGreedy algorithm is better than that of the Chvatal greedy algorithm.

On 10 large-scale instances, we also compares the NNGreedy algorithm and the Chvatal greedy algorithm from the perspectives of the number of instances to find the optimal solution and the quality of the solution:

1. Comparison of the number of instances for obtaining the optimal solution: NNGreedy obtains the optimal solution on one instance, and the Chvatal greedy algorithm does not obtain the optimal solution on any instance.

2. Comparison of solution Quality: On 8 instances, the solution obtained by NNGreedy is better than the solution obtained by the Chvatal greedy algorithm, and on 2 instances, the solution obtained by NNGreedy is inferior to the Chvatal greedy algorithm.

Table 3 shows the specific solution quality comparison of the two greedy algorithms on 10 large-scale instances. The experimental results show that the solution quality of the NNgreedy algorithm is better than that of the Chvatal greedy algorithm.

Although the NNGreedy algorithm is slow to solve due to embedding training, it is acceptable, and the iteration time of each round is about 0.01s to 0.1s

Table 3: Comparison of solution quality on big scale instances

| Instance | Optimal Obj | Chvatal Greedy | | NNGreedy | |
|---|---|---|---|---|---|
| | | Objective Value | Gap | Objective Value | Gap |
| I50_1 | 1559 | 1742 | 11.73% | **1725** | **10.65%** |
| I50_2 | 2206 | 2452 | 11.15% | **2331** | **5.67%** |
| I50_3 | 1221 | 1899 | 55.53% | **1760** | **44.14%** |
| I50_4 | 2548 | 3203 | 25.71% | **3067** | **20.37%** |
| I50_5 | 2315 | 2417 | 4.41% | **2408** | **4.02%** |
| I50_6 | 1965 | **2294** | **16.74%** | 2298 | 16.95% |
| I50_7 | 1933 | 2128 | 9.83% | **2030** | **5.02%** |
| I50_8 | 1975 | 2101 | 6.28% | **2013** | **1.92%** |
| I50_9 | 2105 | **2439** | **15.87%** | 2525 | 19.95% |
| I50_10 | 1630 | 1888 | 15.83% | **1630** | **0** |

# 6 CONCLUSION AND DISCUSSIONS

Aiming at the WSCP, this paper proposes an optimal sub-problem estimation algorithm NNVal and a solving algorithm NNGreeny. Experiments show that the NNVal algorithm and NNGreedy algorithm based on deep learning can obtain better solutions than the Chvatal greedy algorithm based on human experience. The basic idea of our approach can be readily extended without significant modification to design greedy algorithm for other NP-hard problems.

Although NNGreedy is sufficient for neural network training to converge on the corresponding SCP, there is still a certain time gap from commercial solvers. In addition, the hyperparameter setting of the iterative algorithm is still an issue. For the WSCP of the same scale, under the same parameters, the time for the training to converge to the output optimal solution is still not stable.

This paper is an attempt to use deep learning technology to assist algorithm design: the greedy algorithm proposed in this paper does not rely on human experience to design greedy rules, but learns greedy rules, which is an intelligent search algorithm and can help to overcome the inadequacy of human experience. We believe that data-driven algorithm design will be a new hot topic.

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

## A  LOSS FUNCTION FOR ILP

The neural network model is denoted as $g_\theta$ where $\theta$ is the parameters of the neural network. We first need $g_\theta$ satisfy the state-transition equation as follows,

$$g_\theta(A', b', c') = \min\left(g_\theta(A'', b', c''), c_{k+1} + g_\theta(A'', b' - \alpha_{k+1}, c'')\right) \tag{12}$$

We define the following variables:

$$
\begin{aligned}
\text{LHS} = g_\theta(s_k) &= g_\theta(A', b', c') \\
\text{RHS1} = g_\theta(s_k, x_{k+1} = 1) &= c_{k+1} + g_\theta(A'', b' - \alpha_{k+1}, c'') \\
\text{RHS0} = g_\theta(s_k, x_{k+1} = 0) &= g_\theta(A'', b', c'')
\end{aligned}
\tag{13}
$$

We want $g_\theta$ to satisfy the state-transition equation 14:

$$
\text{LHS} = \begin{cases}
\min(\text{RHS1}, \text{RHS0}) & x_{k+1} \text{ is free} \\
\text{RHS1} & x_{k+1} = 0 \text{ is infeasible} \\
\text{RHS0} & x_{k+1} = 1 \text{ is infeasible}
\end{cases}
\tag{14}
$$

There is an important fact that we have defined $f(A, b, c) = +\infty$ when $x_i = 0 \text{ or } 1$ has no feasible solution. This constraint should be taken into the loss function as well. Therefore, the loss function (for one-step state-transition) can be designed as follows:

$$
loss(\theta) = \begin{cases}
(\text{LHS} - \min(\text{RHS1}, \text{RHS0}))^2 & x_{k+1} \text{ is free} \\
(\text{LHS} - \text{RHS1})^2 + \text{ReLU}(\text{RHS1} - \text{RHS0}) & x_{k+1} = 0 \text{ is infeasible} \\
(\text{LHS} - \text{RHS0})^2 + \text{ReLU}(\text{RHS0} - \text{RHS1}) & x_{k+1} = 1 \text{ is infeasible}
\end{cases}
\tag{15}
$$

The purpose of the ReLU item is to make the output of $g_\theta$ large enough when the sub-problem is not feasible. In fact, in one-step state transition, $g_\theta$ can obtain the correct optimal solution as long as the output value of $g_\theta$ in the infeasible state is greater than the one in the feasible. When the training is over, the optimal solution can be obtained according to the equation 16 in each step.

$$
x_{k+1} = \begin{cases}
0 & g_\theta(A'', b', c'') \leq c_{k+1} + g_\theta(A'', b' - \alpha_{k+1}, c'') \\
1 & g_\theta(A'', b', c'') > c_{k+1} + g_\theta(A'', b' - \alpha_{k+1}, c'')
\end{cases}
\tag{16}
$$

## B  TRAINING PROCESS

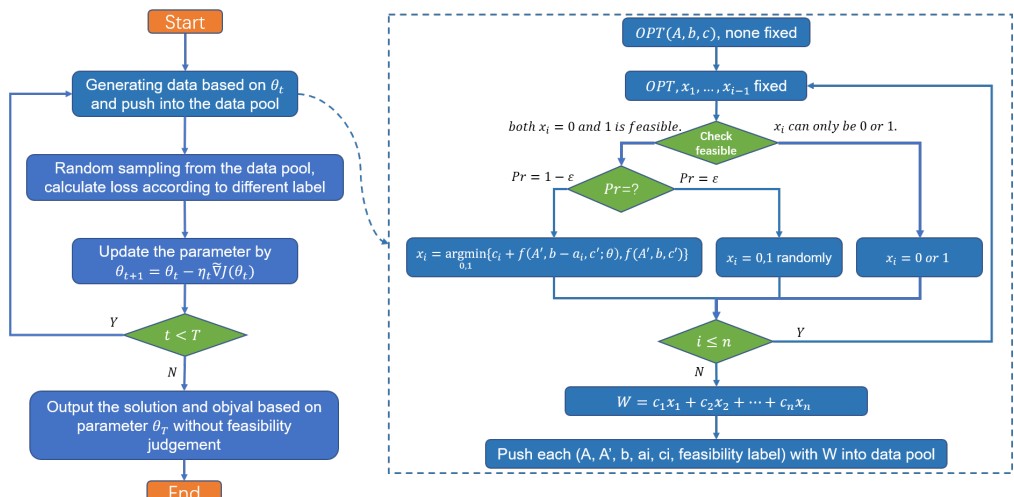

Figure 3: Training process used in NNVal V1

In order to understand the training process of the algorithm, we provide the algorithm flowchart and pseudocode in this section. For algorithm NNVal V1, the flowchart is Figure 3 and the pseudocode is Algorithm 3 and for algorithm NNVal V2, the flowchart is Figure 4 and the pseudocode is Algorithm 4.

---

**Algorithm 3** Training Algorithm for NNVal V1

---

1: Initialize $g_\theta$, data pool D, exploration rate $\epsilon$, learning rate $\eta$ and max iteration number $T$, $t \leftarrow 1$
2: **while** $t < T$ **do**
3:   $Trj, sol \leftarrow$ GENERATETRAJECTORY$(g_\theta, \boldsymbol{A}, \boldsymbol{c})$
4:   Push $Trj$ with weight $\boldsymbol{c}^T \cdot sol$ into data pool D
5:   Generate a batch training data B by weighted sampling in D
6:   Calculate $\mathcal{L}(\theta) = \sum_B L(\theta)$
7:   $\theta \leftarrow \theta - \eta \nabla \mathcal{L}(\theta)$
8:   Evaluate $g_\theta$
9: **end while**
10: **function** GENERATETRAJECTORY$(g, \boldsymbol{A}, \boldsymbol{c})$
11:   $m, n \leftarrow \boldsymbol{A}.shape, \boldsymbol{b}_{tmp} \leftarrow \{1, ..., 1\}^T$
12:   Initialize $sol, Trj$
13:   **for** $i = 1 \rightarrow n$ **do**
14:    $\alpha_i \leftarrow$ the $i$-th column of $\boldsymbol{A}$, $c_i \leftarrow$ the $i$-th number of $\boldsymbol{c}$
15:    $\boldsymbol{A}_0 \leftarrow \boldsymbol{A}$, the $i$-th column of $\boldsymbol{A} \leftarrow 0$
16:    $J_1 \leftarrow$ CHECKFEASIBLE$(\boldsymbol{A}, \boldsymbol{b}_{tmp} - \alpha_i)$, $J_0 \leftarrow$ CHECKFEASIBLE$(\boldsymbol{A}, \boldsymbol{b}_{tmp})$
17:    $R_1 \leftarrow g(\boldsymbol{A}, \boldsymbol{b}_{tmp} - \alpha_i) + c_i$, $R_0 \leftarrow g(\boldsymbol{A}, \boldsymbol{b}_{tmp})$
18:    According to $J_1, J_0, R_1, R_0$, calculate $x_i$ with exploration rate $\epsilon$, update $\boldsymbol{b}_{tmp}$, push $x_i$ into $sol$, push $(\boldsymbol{A}_0, \boldsymbol{A}, \boldsymbol{b}_{tmp}, \alpha_i, c_i)$ into $Trj$
19:   **end for**
20:   **return** $Trj, sol$
21: **end function**
22: **function** CHECKFEASIBLE$(\boldsymbol{A}, \boldsymbol{b})$
23:   **if** $\boldsymbol{A}\boldsymbol{x} \geq \boldsymbol{b}$ is feasible **then**
24:    **return** True
25:   **else**
26:    **return** False
27:   **end if**
28: **end function**

---

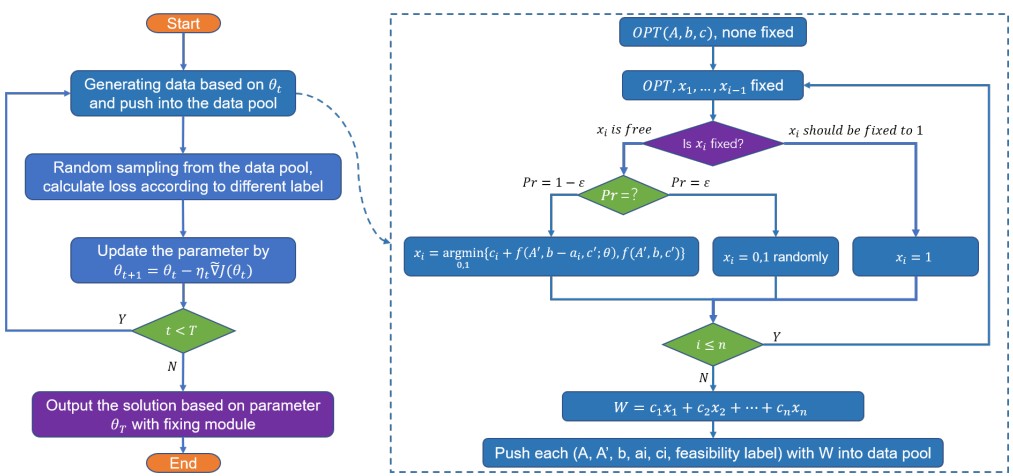

Figure 4: Training process used in NNVal V2

---

**Algorithm 4** Training Algorithm for NNVal V2

---

1: Initialize $g_\theta$, data pool D, exploration rate $\epsilon$, learning rate $\eta$ and max iteration number $T$, $t \leftarrow 1$
2: **while** $t < T$ **do**
3:  $Trj, sol \leftarrow$ GENERATETRAJECTORY$(g_\theta, \boldsymbol{A}, \boldsymbol{c})$
4:  Push $Trj$ with weight $\boldsymbol{c}^T \cdot sol$ into data pool D
5:  Generate a batch training data B by weighted sampling in D
6:  Calculate $\mathcal{L}(\theta) = \sum_B L(\theta)$
7:  $\theta \leftarrow \theta - \eta \nabla \mathcal{L}(\theta)$
8:  Evaluate $g_\theta$
9: **end while**
10: **function** GENERATETRAJECTORY$(g, \boldsymbol{A}, \boldsymbol{c})$
11:  $m, n \leftarrow A.shape, \boldsymbol{b}_{tmp} \leftarrow \{1, ..., 1\}^T$, initialize $sol, Trj$
12:  **for** $i = 1 \rightarrow n$ **do**
13:    $\alpha_i \leftarrow$ the $i$-th column of $\boldsymbol{A}$, $c_i \leftarrow$ the $i$-th number of $\boldsymbol{c}$, $\boldsymbol{A}_0 \leftarrow A$,
14:    $label, fix\_idxs \leftarrow$ CHECKFIXING$(\boldsymbol{A}, \boldsymbol{b}_{tmp}, i)$
15:    **if** $label == True$ **then**
16:      Push $x_i = 1$ into $sol$, the $i$-th column of $\boldsymbol{A} \leftarrow 0$
17:      Push $(\boldsymbol{A}_0, \boldsymbol{A}, \boldsymbol{b}_{tmp}, \alpha_i, c_i)$ into $Trj$, $\boldsymbol{b}_{tmp} \leftarrow \boldsymbol{b}_{tmp} - \alpha_i$
18:    **else**
19:      The $i$-th column of $\boldsymbol{A} \leftarrow 0$, push $(\boldsymbol{A}_0, \boldsymbol{A}, \boldsymbol{b}_{tmp}, \alpha_i, c_i)$ into $Trj$
20:      $R_1 \leftarrow g(\boldsymbol{A}, \boldsymbol{b}_{tmp} - \alpha_i) + c_i$, $R_0 \leftarrow g(\boldsymbol{A}, \boldsymbol{b}_{tmp})$
21:      According to $R_1, R_0$, calculate $x_i$ with exploration rate $\epsilon$, update $b_{tmp}$, push $x_i$ into
  $sol$
22:    **end if**
23:  **end for**
24:  **return** $Trj, sol$
25: **end function**
26: **function** CHECKFIXING$(\boldsymbol{A}, \boldsymbol{b}, idx)$
27:  $m, n \leftarrow \boldsymbol{A}.shape, label \leftarrow False$, initialize $fix\_idxs$
28:  **for** $i = 1 \rightarrow m$ **and** $b[i] < 1$ **do**
29:    Calculate $cnt$, which is the number of 1 in $\boldsymbol{A}[i]$, $pos$ means a position of 1
30:    **if** $cnt == 1$ **then**
31:      Push $pos$ into $fix\_idxs$
32:    **end if**
33:  **end for**
34:  **if** $idx$ in $fix\_idxs$ **then**
35:    $label \leftarrow True$
36:  **end if**
37:  **return** $label, fix\_idxs$
38: **end function**

---

