# OpenReview forum: "AIA: learn to design greedy algorithm for NP-complete problems using neural networks"
_ICLR.cc/2023/Conference — Submitted to ICLR 2023_

### Official Review · Reviewer_CYCn · 2022-10-22

**Confidence:** 4
**Clarity, Quality, Novelty And Reproducibility:** Overall the paper is understandable. …
**Correctness:** 2
**Technical Novelty And Significance:** 2
**Empirical Novelty And Significance:** 2
**Recommendation:** 3

**Strength And Weaknesses:**

Strengths:

The results presented show seemingly significant improvements.

Weaknesses:

1. There is only one experiment conducted on a random dataset. Also the compared method is only a classical method. There exists some other heuristic methods which are mentioned but not compared. Thus the improvement of the method is not quite convincing.

2. The meta-heuristic approaches are mentioned in the introduction without any citations.

3. The design of the neural network seems trivial. The loss function is designed but why does it work well is not estabished.



**Summary Of The Paper:**

This paper proposes a neural network based machine learning method for deciding binary variables at each search step. Some design decisions such as layers of the neural network, the loss functions and etc are made. Some numerical results to demonstrate the performance of the proposed method are given.

**Summary Of The Review:**

Overall, the contribution of this paper is marginal. The numerical experiments are not carried out extensively so the improvement of the proposed method is hard to justify.

---

### Official Review · Reviewer_3jeR · 2022-10-24

**Confidence:** 4
**Correctness:** 2
**Technical Novelty And Significance:** 3
**Empirical Novelty And Significance:** 3
**Recommendation:** 3

**Clarity, Quality, Novelty And Reproducibility:**

It is difficult to evaluate this paper due to omissions and missing information.

The authors further claim that "The basic idea of our approach can be readily extended without significant modification to design greedy algorithm for other NP-hard problems." but this is never addressed.


Please give details on  how you map WSCP from a graph problem to an optimization problem as defined. Is this meant to be a dynamic programming formulation? You MUST give references to previous task formulations.

In sec. 4, it is confusing what you mean by "We use a neural network to score each sub-problem and guide multi-step decision-making according to equation 3 instead of predicting the solution to the sub-problem."

State precisely what you mean by "score each sub-problem" and "predicting the solution to the sub-problem".

In sec. 4.2 you also state "accurately predict the target": what does this mean precisely?

State precisely how the feasibility of the subproblem is computed.

**Strength And Weaknesses:**

Strengths:

This is an important research area, and the paper has many aspects of novelty.

Weakesses:

The approach is interesting, but the technical details of the paper are so poor that it is very hard to understand how the algorithm works. Many critical concepts are not defined.

**Summary Of The Paper:**

This paper proposes an approach, called AIA, to learn algorithm design with the aid of neural networks. We consider the minimum weighted set cover problem (WSCP), one of the NP-hard problems, as an representative example.






**Summary Of The Review:**

This is an interesting paper on a critical topic. However, the paper is technically weak and does not allow a reviewer to evaluate it fully.

---

### Official Review · Reviewer_63py · 2022-10-24

**Confidence:** 4
**Correctness:** 2
**Technical Novelty And Significance:** 2
**Empirical Novelty And Significance:** 2
**Recommendation:** 3

**Clarity, Quality, Novelty And Reproducibility:**

The paper is not very clear, and while the particular approach seems novel, not enough details is given to reproduce the results.

**Strength And Weaknesses:**

+ interesting approach to a classic NP-complete problem

- paper not well written and lacks detail to understand what exactly is being proposed
- experimental evaluation small-scale and compares to only one method
- many technical details unclear

**Summary Of The Paper:**

The paper proposes a neural-network-based approach to learn greedy heuristics for the WSCP. The authors describe their method and evaluate it empirically, comparing it to a hand-crafted heuristic.

**Summary Of The Review:**

The paper investigates an interesting problem, but lacks detail and clarity. This starts with the abstract, which mentions that the loss function should satisfy the Bellman-Ford equation, but no detail of this is given in the paper itself (neither the equation, nor how the loss function satisfies it. The description of the method uses symbols that are not explained. The problem itself is never specified, at least it is not obvious how the state and a solution to the WSCP relate. The architecture for the neural network to solve the problem is simply given, not justified or explained -- why this particular architecture? Did you compare to any other architecture candidates? The authors give two versions of their neural network, the second improving upon the first, but this is not empirically evaluated. It is unclear why two methods are given if only one is evaluated.

The authors claim that the experimental evaluation considers both execution time and solution quality, but execution times are not reported, only at a very high level that does not allow the reader to compare to the other method. The only other method the authors compare to is from 1979, and the proposed new method does not improve in every case. Only two small sets of relatively uniform instances are considered. It is unclear how the accuracy of the estimation order was evaluated; accuracy seems unsuitable in this case and a rank measure should be used. The authors do not say how they trained their neural network. Based on the description given in the paper, it is impossible to reproduce the results.

No reference is given for the Chvatal algorithm, which is referred to extensively in the paper. There is no space before references. The authors in a reference are sometimes referred to by their last names, sometimes by their first names. There are typos, grammatical mistakes, and formatting mistakes throughout the paper.

---

### Official Review · Reviewer_Vs8A · 2022-10-29

**Confidence:** 3
**Correctness:** 2
**Technical Novelty And Significance:** 2
**Empirical Novelty And Significance:** 2
**Recommendation:** 5

**Clarity, Quality, Novelty And Reproducibility:**

About Novelty:
- Not really novel. Solving an NP-Complete problem using a Neural Network by leveraging a recurrence relation has been studied extensively. As stated in the paper, previous works include using Reinforcement Learning and Graph Neural Networks.
- Although the Loss function for this specific scenario is never used before, I feel that this loss function is not extremely novel.

About Organization and Presentation:
- Clear, I like how this paper was organized, it is clear, and somewhat concise in what it is trying to do.
- Each section is really plain and simple to understand. Figures 1 & 2 are very clear and are really intuitive as to how the information is reshaped and passed through the MLP.

**Strength And Weaknesses:**

- The Authors contribute to this area by presenting a more or less novel method of leveraging the recurrence relations between each state. Making an NN estimate the relation between each state.
- The authors claim that the NN can aid the researchers' design of an algorithm, implying that this is not an end-to-end model. But in the 4.5 NNGreedy algorithm section, essentially, it seems that the algorithm is just deciding whether a node should or should not be included, based on the state information of the node. Which is more or less an end-to-end model.
- A lot of work on tackling NP-Complete problems using Neural structures has been presented already, what is the key difference between using the plain MLP and say a GNN? Like in this work https://arxiv.org/abs/1809.02721 for example.
- If one is trying to leverage the recurrence relations between each state, would it be more intuitive to just use an RNN or a GCNN? What is the advantage of using the plain MLP and not his alternatives?
- Baseline method is limited to a human designed heuristic algorithm. It is not enough to convince that the algorithm could outperform existing methods. A baseline of optimal solutions should be added as well.

**Summary Of The Paper:**

The paper tackles the problem of the minimum weighted set problem by using a Neural Network. They leverage the recurrence relationship of the state-transition equation, and instead of making the NN learn directly the value of the sub-problems, they make the NN learn the relationship between each state. The model provides a relatively good result, that beats the Chvatal Greedy method in most instances.

**Summary Of The Review:**

About Challenge:
-The authors claim that they overcame the main challenge by overcoming the idea of fitting the recurrent relation with a neural network. Which is unfeasible due to the neural network being very sensitive to the change in input.

---

### Decision · Program_Chairs · 2023-01-20

**Decision:**

Reject

**Justification For Why Not Higher Score:**

N/A

**Justification For Why Not Lower Score:**

N/A

**Metareview: Summary, Strengths And Weaknesses:**

This paper considers the problem of solving minimum weighted set cover using neural networks as greedy policies for sequential decision-making about which objects to include or not (binary decision). The domain knowledge about Bellman-Ford constraint is used as part of the loss function to train the neural network.

There is a consensus among the reviewers' that the paper is lacking in the following aspects and is not ready for publication.
- There is a large body of work on using neural networks to solve combinatorial optimization problems. The technical novelty w.r.t to this literature is limited. Authors' are encouraged to improve the exposition to better highlight the novel aspects and how the ideas are more useful for problems beyond the specific combinatorial optimization problem studied in this paper.
- The paper is not well-written and has a lot of missing information and references which hinders both readability and evaluation.
- Experimental evaluation is lacking in many ways.

For all the above reasons, I'm recommending to reject this paper and encourage the authors' to improve the paper based on the feedback from reviewers'.